# Effect of Cryotherapy Temperature on the Extension Performance of Healthy Adults’ Legs

**DOI:** 10.3390/biology10070591

**Published:** 2021-06-26

**Authors:** Yichen Lu, Yuqi He, Shanshan Ying, Qiaojun Wang, Jianshe Li

**Affiliations:** 1Zhejiang Pharmaceutical College, Ningbo 315100, China; luyc@mail.zjpc.net.cn; 2Research Academy of Grand Health, Ningbo University, Ningbo 315211, China; 1811042041@nbu.edu.cn (Y.H.); lijianshe@nbu.edu.cn (J.L.); 3Faculty of Sports Science, Ningbo University, Ningbo 315211, China; wangqiaojun@nbu.edu.cn; 4Faculty of Engineering, University of Pannonia, 8200 Veszprém, Hungary

**Keywords:** cryotherapy, temperature, extension performance, peak power, peak force

## Abstract

**Simple Summary:**

The aim of this study was to investigate the effect of different temperature cryotherapy on leg extension performance. A short period of cold treatment at 0 °C may increase the leg extension ability; however, further lower cold treatment temperatures may cause a decrease in the stretching speed. This may indicate that 0 °C cold therapy for a short period of time is more suitable than −5 °C cold therapy for athletes to recover during the interval within a competition.

**Abstract:**

Cryotherapy as a measurement of sports recovery and treatment has been utilized by more and more athletes and coaches. The aim of this study was to investigate the effect of different temperature cryotherapies on leg extension performance. Thirty-one male participants from a University volunteered to join two different temperature cryotherapies. The peak power and peak force of right leg extension performance of each participant was measured by Keiser, before and after cryotherapy, respectively. The results of this study show that there was a significant difference in peak power (t = −3.203, *p* value = 0.003) and peak force (t = −4.662, *p* value = 0) before and after 0 °C cryotherapy. In addition, there was a significant difference in peak force (t = −4.36, *p* value = 0) before and after −5 °C cryotherapy. Besides, the changing rates of peak power (3.03%) and peak force (11.51%) in the 0 °C group were higher than those of peak power (2.80%), as well as peak force (7.45%), in the −5 °C group. The PF in the 0 °C and −5 °C groups both significantly increased after cryotherapy. Peak power significantly increased after 0 °C cryotherapy, but did not significantly increase after −5 °C cryotherapy. The changing rates of peak power and peak force in the 0 °C group were both higher than the −5 °C group. A short period of cold treatment at 0 °C may increase the leg extension ability. A 0 °C cold therapy for a short period of time is more suitable than a −5 °C one for athletes to recover during the interval within a competition. Athlete and coach could choose an appropriate temperature to help increase performance of physical level and recovery.

## 1. Introduction

As a new measure of sports recovery and treatment [1,2,3], cryotherapy is being accepted by more and more athletes and coaches. Although cryotherapy has been extensively studied as a practical measure of recovery, it has received less attention. The functions of cryotherapy include inflammation control, edema reduction and pain reduction [1,3,4,5,6]; it helps the post-traumatic recovery process by lowering muscle temperature, reducing pain and muscle cramps and reducing inflammatory processes. Although cryotherapy has played a positive role in rehabilitation, several studies have demonstrated the adverse effects of cold therapy on exercise [7,8,9,10,11]. Reduced muscular torque, inappropriate peripheric feedback for proprioception, as well as changes of the biomechanical properties of articulations, are the negative influences of cryotherapy on motor control [11,12]. This evidence indicates that the study of cold therapy has received extensive attention from scholars and that further research is needed to reveal the underlying mechanism of cold therapy. The study of leg extension has received a lot of attention. Increased leg extension increases propulsion in a gait cycle [13]. Taichi et al. (2005) investigated the effects of static stretching for 30 s and dynamic stretching on the extension performance of legs [14]. Bassey et al. (1990) investigated a new method to measure power output in a single leg extension [15]. Exploring the effect of cold therapy on the extension performance of the dominant leg may help to understand the mechanism of muscle response to cryotherapy.

Cold therapy is a cooling process that involves the gradual penetration through the surface skin muscle to deep within the skin. Due to the body’s temperature self-regulation mechanism, subcutaneous tissues begin to warm almost immediately after cryotherapy treatment, whereas deeper tissues continue to cool after removing the cooling agent [16,17]. The rapid contraction of deep muscles may be affected due to the cooling effect. These changes in intensity are attributed to a decrease in the hydrolysis of 5′-adenosine triphosphate, which results in a decrease in the maximum rate of muscle contraction [17]. Several studies investigated the effect of cryotherapy on the lower limb ability. Schmid et al. (2010) reported that the performance of vertical jump showed reduction after cryotherapy [9]. Wang et al. (2010) reported similar results [10]. However, Rowsell et al. (2009) [18] and Antonio et al. (2011) [13] reported on the performance of jump after cryotherapy by cold water at 10 °C and water between 34 °C and 35 °C and the results show that there was no significant difference between the two groups. William et al. (1998) investigated information on the temperature changes of the human leg during and after two 20-minute cryotherapy methods, showing that there was no significant difference in the intramuscular temperatures [16]. Further, the effect of cryotherapy on muscle activity was investigated in recent years. The peak power (PP) and peak force (PF) of lower limbs are an important index to reflect human physical functions and the impact of cryotherapy on functional activities, which are more related to the specific needs of athletes [16]. Costello et al. (2012) investigated the muscle force recovery and proprioceptive function after −110 °C whole-body cryotherapy (WBC); the position sense, maximal voluntary and force proprioception of the knee joint were not affected by WBC [19].

The evidence suggests that the study on the effects of cryotherapy on muscle activity in the human lower extremities is significant. There are several past studies investigating the operation methods and treatment time of cryotherapy. To the temperature of the frozen treatment, research has not yet paid too much attention. However, as the main factor influencing the effects of cryotherapy, studying the influence of the temperature on freezing treatment effects is conducive to further explore the internal mechanism of cryotherapy. Besides, the study by Hohenauer et al. (2015) listed 19 cold therapy literatures before 2015 and showed that there were 16 experimental studies involving cold water immersion (CWI) for which the intervention temperature range was 3–15 °C [20]. Generally, the mixture of ice and water is 0–4 °C; previous studies have shown that, in cold treatment methods for direct contact of solid and liquid with skin, the treatment temperatures of ice, water, or a mixture of ice and water were all above or near 0 °C. To our knowledge, we did not see any studies on CWI and ice pack cryotherapy using temperatures below 0 °C. In this study, we compared the cryotherapy temperatures of 0 °C and −5 °C. Previous studies have focused on the recovery of athletes after fatigue and the cryotherapy of patients, but it is also meaningful to study the effects of cryotherapy on healthy people. By exploring the influence of cold therapy at different temperatures on the stretching ability of the dominant leg in healthy people, the internal mechanism of cryotherapy can be further understood. A number of studies evaluated the output of the lower extremities through the performance of the jumping movement [9,10]. In fact, this approach has some limitations, namely, it cannot be evaluated separately from explosive power and the ability to overcome resistance. Through the observation of the PP and PF indexes, the influence of cryotherapy on explosive force and the ability to overcome resistance can be understood on a deeper level. Therefore, the aim of this study is to investigate the effects of 0 °C and −5 °C temperature cryotherapy on the dominant leg extension performance via measurement of the mechanical power output. It is hypothesized that, with a decrease in cryotherapy temperature, the participants’ dominant leg extension performance will exhibit a different performance level of PP and PF compared with warmer cryotherapy temperatures.

## 2. Materials and Methods

### 2.1. Participants

Thirty-one male participants volunteered to join this study. Detailed demographic information is showed in Table 1. The ethics committee of Ningbo University approved this study. All the participants were from Ningbo University. All participants were asked to measure the experimental indexes on the right leg. There was no injury or disorders in the lower limb prior to the six-month period before this study. Each participant was informed about the objective and test details before the official experiment. Besides, participants could quit the experimental anytime.

### 2.2. Experimental Design

The experimental place was set at the Research Academy of Grand Health in the University. In this study, the fast-stretching ability was defined as peak power (W). The maximum force to overcome resistance of the leg stretching ability was defined as peak force (Lb). The peak power and peak force of the leg were measured by Keiser (Fresno, CA, USA, 002521PP).

All participants were instructed to control the position between human body and machine during tests. According to the length of the participant’s leg, the position of the seat was adjusted to ensure the angle between knee and hip joint was 90 degrees before the activity of stretch. The hands and the other leg of the participant were asked to be put in a fixed position to ensure the same test conditions for each participant. Before the test started, all participants could practice and familiarize themselves with the machine, as well as the environment, for enough time. Then the participants were asked to receive cryotherapy. All participants joined two experiment sessions, including one test at 0 °C and the other one at −5 °C. There were two weekends between the two tests to avoid the influence. The cryotherapy intervention at 0 °C was arranged in the first week, followed by the cryotherapy intervention at −5 °C, two weeks later. A cryotherapy machine (Chenhui Medical, Suzhou, China) was used to provide a cold interface at the temperatures of 0 °C and −5 °C. The time of cryotherapy was set at 5 min for each group. During the official experiment, at first, the peak power and peak force of each participant during leg extension were measured as a basic criterion. After the subjects completed the cold therapy intervention, the PP and PF of leg extension were measured 5 times with rest periods of 15 s between trials. The mean of the 2 maximum values in the 5 trials was taken as the leg extension PP and PF of each subject [14].

### 2.3. Instrumentation

As shown in Figure 1 and Figure 2, Keiser (USA, 002521PP) was used in this study to measure PP and PF of the subjects’ leg extension ability. Meanwhile, participants were cryotherapy treated with a cryotherapy machine (Chenhui Medical, Suzhou, China). The cryotherapy treatment equipment used in this study was cooled by a compressor and R134A tetrafluoroethane and an antifreeze fluid in the bladder were in contact with the skin. The lowest temperature that can be controlled by the device is −5 °C and the maximum time of cryotherapy is 30 min. The device was refrigerated by a compressor, which made the antifreeze reach a constant −5 °C and the heat was exchanged with the skin through the bladder.

### 2.4. Data Statistics and Analysis

In the PASS16.0 software, we performed the following operation to finish the sample size calculation: select means, two independent means; test (inequality), test for two means (two-sample *t*-tests) (differences). The sample size was calculated as 31 (effect size = 0.669, power = 0.95). The SPSS 21.0 statistical software (SPSS Inc., Chicago, IL, USA) was used for the statistical analysis of the collected data, which were expressed in the form of mean ± SD. The boxplot and Shapiro–Wilk test were used to check normal distribution. A paired-samples *t*-test was used to compare the peak power and peak force between the 0 °C and −5 °C groups. Significance level was set at *p* < 0.05.

## 3. Results

### 3.1. Experimental One-Effect of PP and PF in 0 °C Cryotherapy

Comparisons of PP and PF between before and after 0 °C cryotherapy are displayed in Table 2 and Figure 3. There was a significant difference in PP (t = −3.203, *p* value = 0.003) and PF (t = −4.662, *p* value = 0) between before and after 0 °C cryotherapy. That means that the Δ(After–Before) of PP and PF was statistically significant.

### 3.2. Experimental Two-Effect of PP and PF in −5 °C Cryotherapy

Comparisons of PP and PF between before and after −5 °C cryotherapy are displayed in Table 3 and Figure 3. There was a significant difference in PF (t = −4.36, *p* value = 0) between before and after −5 °C cryotherapy. That means that the Δ(After–Before) of PF was statistically significant. However, there was no significant difference in PP (t = −0.19, *p* value = 0.85) between before and after −5 °C cryotherapy.

### 3.3. Changing Rates of PP and PF between 0 °C and −5 °C Cryotherapy

Comparisons of changing rates of PP and PF between 0 °C and −5 °C cryotherapy are displayed in Figure 4. The changing rate of PP (3.03%) and PF (11.51%) in the 0 °C group were higher than PP (2.80%), as well as PF (7.45%), in the −5 °C group.

## 4. Discussion

The aim of this study was to investigate the effects of different temperature cryotherapies on leg extension performance. The key findings of this study are: (1) the PFs in the 0 °C and −5 °C groups both significantly increased after cryotherapy; (2) PP significantly increased after 0 °C cryotherapy, but did not significantly increased after −5 °C cryotherapy; (3) the changing rates of PP and PF in the 0 °C group were both higher than the −5 °C group.

PP and PF are the main indexes to evaluate the performance of leg extension ability. In this study, the performance of PP and PF of leg extension was improved after cryotherapy, but all results show that 0 °C cryotherapy has a better influence on leg extension ability compared with −5 °C cryotherapy. Time and temperature, as the main factors of cryotherapy, influence the result of performance recovery. Limited cryotherapy time and temperature affect the degree of cooling of shallow and deep tissues, leading to a cold therapy effect that is not ideal. This is because cold therapy is a cooling process that involves the gradual penetration through the surface skin muscle to deep within the skin and is also affected by the heat preservation reduction effect caused by different thickness of sebum [21], as well as the human circulatory system influence on temperature due to stress [22], etc. Athletes are usually treated with cryotherapy immediately after physical injury [23]. After undergoing cryotherapy for acute knee injuries, athletes can return to training or competition [24]. Abundant physiological and clinical evidence show that cold compresses can effectively reduce nerve conduction velocity [25], muscle power and muscle force generation [26]. Nerve conduction velocity slows down by 1.5–2 m/s for every 1 °C drop in skin temperature [27] and spindle discharge rate of muscle decreases by 1–3 pulses per second for each 1 °C drop in muscle temperature [28]. Hence, we can speculate the main reason for the result of this study, which is that the performance under 0 °C cryotherapy is better than −5 °C cryotherapy, was that the chilling effect reached deeper and affected the deep muscle under −5 °C cryotherapy. That resulted in a severe decrease in nerve conduction velocity, compared with 0 °C cryotherapy. Point et al.’s (2018) estimated muscle stiffness using ultrasonic shear wave elastography after air-pulsed (−30 °C) cryotherapy; the results show that cryotherapy induced an increase in muscle stiffness [29]. There is evidence that cryotherapy impairs the function of sensory receptors and motor nerve conduction [30], possibly related to a reduction in myosin ATPase activity, leading to a reduction in the ability to generate force [31]. Lower body temperature leads to lower electrical currents in the membranes of the nerves, which, in turn, prolongs the refractory period of cells. When skin temperature drops by 1 °C, nerve conduction velocity decreases by 0.4 m/s [31]. In addition, cryotherapy may also affect the viscosity of muscles. This does not accord with the results of this study, to a certain extent. In this study, PP showed no significant difference, compared with before cryotherapy in the −5 °C group, but PF showed a significant increase. A possible reason for this is that PF testing was arranged after that of PP in the experiment design; the PP test caused muscle temperature to rise, reduced the viscosity of muscles and resulted in the improved output performance of PF. This may be one of the limitations of this study. From a practical standpoint, cryotherapy can be used during short game breaks or halftime breaks, allowing an appropriate rewarming period before a match to limit the cooling effect and reduce potential functional loss [32].

The effects of cold therapy on physical performance are still controversial. It has been suggested that cooling muscles directly reduces the speed of muscle contractions and the ability of athletes to produce force after cryotherapy. Anthony (1987) compared the peak force and peak power of four different temperature cryotherapy groups and a normal rest group. The results of their study show that peak force and peak power increased by 11% in a warmer temperature group, compared with the normal rest group. Moreover, the peak force and peak power in the coldest cryotherapy group showed a 21% decrease, compared with the normal rest group [26]. Besides, Bergh et al. (1979) investigated the effects of muscles and temperatures on dynamic and static maximal strengths. They reported that temperature has a significant influence on dynamic exercise. Moreover, possibly because of the lower nerve impulse frequency, it is more difficult to activate the motor units during a short time interval under a low temperature environment. Furthermore, the speed of chemical reactions is decreased at low temperatures [33]. Thus, the breaking and formation of the cross-bridges may be considerably delayed at low temperatures, resulting in a slower rate of tension development [34,35]. However, Hopkins (2006) investigated the muscle recruitment changes and knee joint function after joint effusion and subsequent joint cryotherapy. The results of their study show that the knee anterior joint reaction of the cryotherapy group was greater than the control and normal groups [35]. Moreover, the study by Kimberly et al. (2012) investigated a performance test of 22 collegiate football athletes within 48 h after cold water immersion [36]. Their results show that cryotherapy does not influence the subsequent physical performance estimates. In this study, the PP after −5 °C cryotherapy showed no significant differences, compared with before therapy; however, the PP after 0 °C cryotherapy showed to be significantly greater than before cryotherapy. We can speculate that, compared with force gathering ability, nerve conduction velocity is influenced by temperature to a large extent. Moreover, maybe this reason could explain the result of this study that the PF after −5 °C cryotherapy showed to be significantly greater than before cryotherapy.

There are still some limitations in this study that should be mentioned. Firstly, the electromyographic activity of muscles should be measured and analyzed in future studies; that would provide implications on the microscopic efficacy of cryotherapy. Secondly, there are only two temperature settings in this study; including more temperature intervals in future studies is necessary. Lastly, we mainly investigated the influence of cryotherapy at different temperatures in a short period of time, in the current study, while the effects of cryotherapy duration may show different efficacy, which shall be considered in future studies.

## 5. Conclusions

This study further illustrates the effects of temperature on leg extension performance. A short period of cold treatment at 0 °C may increase the leg extension ability. However, further lower cold treatment temperatures may cause a decrease in the stretching speed. From a practical standpoint, this may indicate that 0 °C cold therapy for a short period of time is more suitable than −5 °C cold therapy for athletes to recover during the interval within a competition. Athlete and coach could choose an appropriate temperature to help increase the performance of physical level and recovery.

## Figures and Tables

**Figure 1 biology-10-00591-f001:**
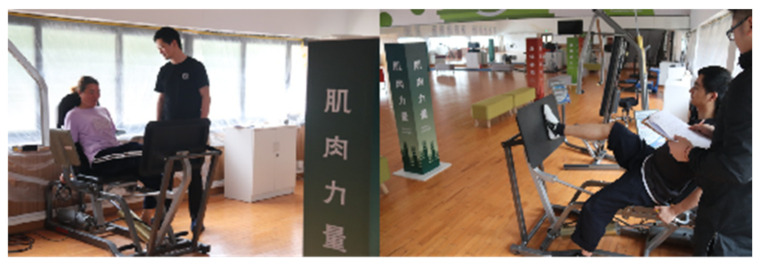
The peak power and peak force tests.

**Figure 2 biology-10-00591-f002:**
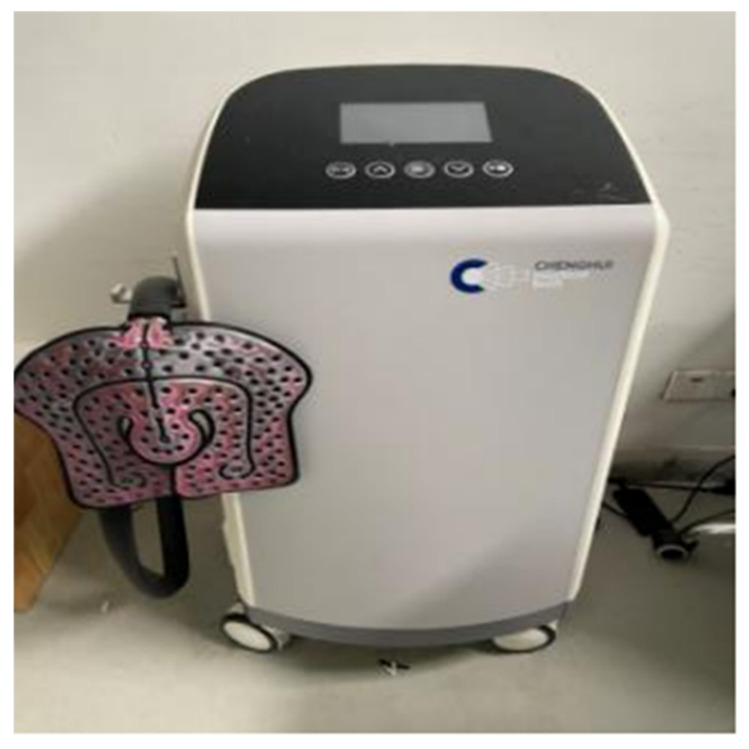
The self-regulating cryotherapy.

**Figure 3 biology-10-00591-f003:**
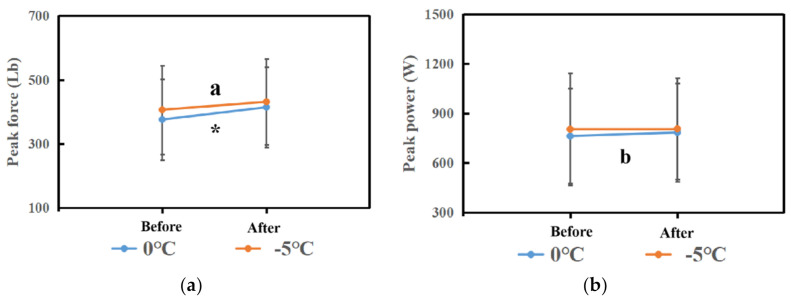
(**a**) The comparison of peak force in each phase between the 0 °C and −5 °C groups. (**b**) The comparison of peak power in each phase between the 0 °C and −5 °C groups. Note: * means that there are significant differences between Before and After in peak force in the 0 °C group. The “a” means that there are significant differences between Before and After in peak force in the −5 °C group. Moreover, the “b” means that there are significant differences between Before and After in peak power in the 0 °C group.

**Figure 4 biology-10-00591-f004:**
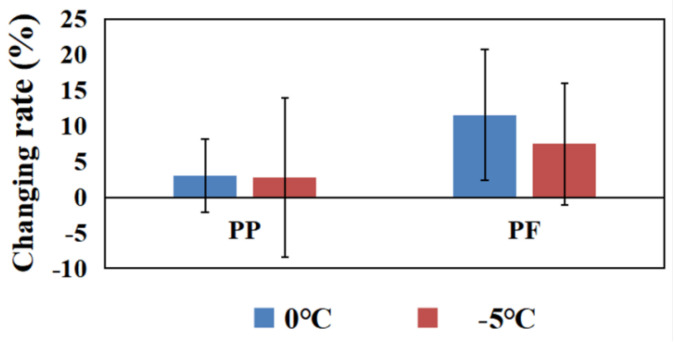
The changing rates of PP and PF in the 0 °C and −5 °C groups. Note: * means that there are significant differences between Before and After in peak power in the 0 °C group.

**Table 1 biology-10-00591-t001:** Population information of the participants (mean ± SD).

	Age (Years)	Height (cm)	Weight (kg)
Participant (*n* = 31)	39.94 ± 12.16	162.59 ± 6.16	60.53 ± 9.96

**Table 2 biology-10-00591-t002:** The information on PP and PF between before and after 0 °C cryotherapy (mean ± SD).

Index	Before	After	*p* Value	*Δ*(After–Before)	95% CI	t	df
PP (W)	762.84 ± 289.33	784.77 ± 298.93	0.003	21.93 ± 9.60 *	−35.92~−7.95	−3.203	30
PF (Lb)	375.90 ± 125.80	414.48 ± 125.75	0	38.58 ± 0.05 *	−55.48~−21.68	−4.662	30

Note: * means that there are significant differences between before and after phases. Before means before cryotherapy, After means after cryotherapy. PP refers to peak power, PF refers to peak force.

**Table 3 biology-10-00591-t003:** The information of PP and PF between before and after −5 °C cryotherapy. (Mean ± SD).

Index	Before	After	*p* Value	*Δ*(After–Before)	95% CI	t	df
PP (W)	804.45 ± 339.57	807.16 ± 307.62	0.85	2.71 ± 31.95	−31.79~26.37	−0.19	30
PF (Lb)	406.58 ± 138.66	431.39 ± 133.99	0	24.81 ± 4.67 *	−36.55~−13.07	−4.36	30

Note: * means that there are significant differences between Before and After phases. Before means before cryotherapy, After means after cryotherapy. PP refers to peak power, PF refers to peak force.

## Data Availability

The data that support the findings of this study are available on rea-sonable request from the corresponding author. The data are not publicly available due to privacy or ethical restrictions.

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
