# Peer review of "Effect of Cryotherapy Temperature on the Extension Performance of Healthy Adults’ Legs"

_biology, 2021, doi:10.3390/biology10070591_

Round 1
Reviewer 1 Report
The aim of this study was to investigate the effect of different temperature cryo-12 therapy on the dominant leg extension performance.
The study presents a well-developed design and has been rigorously conducted. Some considerations are provided in order to improve understanding.
1. Title: refers to the fact that the experiment has been carried out on "the dominant leg", and this aspect is continuously considered throughout the study. My reflection to the authors is whether the consideration of being the "dominant leg" is an important variable, or is it simply to determine a leg in the analysis. In the first case, it should be justified in the study; In the second case, consider not including this aspect in the title, since it leads to the understanding that in the non-dominant leg, the results could be different.
2. Directly related to the previous point, the authors declare that the right leg was considered dominant leg, but the way in which this lateral dominance was determined is not explained. If the choice was determined by left or right, then it does not make sense for the expression "dominant leg" to appear in the title. If the choice was through any test, this test must be explained, since it would be an element that would determine the "object of the study" (that is, on the dominant leg). That said, I suggest to the authors that they reflect on the concept of "dominant leg", since, for example, in a kick at the ball with the foot: is the dominant leg the one that kicks the ball, or is the dominant leg the one that rests on the ground? This problem should be clarified in the paper, since it is an aspect that the authors have included in the title, and on which they refer throughout the work.
3. For example, and finally, in line 281 (conclusion section), it is declared that "A short period of cold treatment at 0 ℃ may increase the extension ability of the dominant leg"; Does this mean that the results may be different on the other leg? If the answer is affirmative, what have been the clear criteria to determine that this leg is the dominant one?
For the rest, congratulate the authors for the work that, although the topic has already been deeply studied, the conditions established in the design will contribute to the increase of knowledge.
Author Response
The aim of this study was to investigate the effect of different temperature cryotherapy on the dominant leg extension performance. The study presents a well-developed design and has been rigorously conducted. Some considerations are provided in order to improve understanding. For the rest, congratulate the authors for the work that, although the topic has already been deeply studied, the conditions established in the design will contribute to the increase of knowledge.
Response: It is a great honor for us to receive your recognition, which greatly encourages our future research work in this field. At the same time, we have further revised the manuscript according to your suggestions, and we fully accept your suggestions and comments. Thank you very much for your valuable suggestions on improving the quality of our manuscripts.
Comment 1.Title: refers to the fact that the experiment has been carried out on "the dominant leg", and this aspect is continuously considered throughout the study. My reflection to the authors is whether the consideration of being the "dominant leg" is an important variable, or is it simply to determine a leg in the analysis. In the first case, it should be justified in the study. In the second case, consider not including this aspect in the title, since it leads to the understanding that in the non-dominant leg, the results could be different.
Response: Thank you very much, we totally agree with your suggestion. In order to make readers better understand the content of our research, we have revised the title and manuscript. The title has been changed to: Effect of cryotherapy temperature on the extension performance in healthy adults leg.
Comment 2. Directly related to the previous point, the authors declare that the right leg was considered dominant leg, but the way in which this lateral dominance was determined is not explained. If the choice was determined by left or right, then it does not make sense for the expression "dominant leg" to appear in the title. If the choice was through any test, this test must be explained, since it would be an element that would determine the "object of the study" (that is, on the dominant leg). That said, I suggest to the authors that they reflect on the concept of "dominant leg", since, for example, in a kick at the ball with the foot: is the dominant leg the one that kicks the ball, or is the dominant leg the one that rests on the ground? This problem should be clarified in the paper, since it is an aspect that the authors have included in the title, and on which they refer throughout the work.
Response: Thank you very much for your suggestions on improving the quality of this study. In fact, the right leg was chosen as the test leg for this study, and we have revised the title and manuscript to avoid possible confusion. So the readers can understand our research more clearly.
Comment 3. For example, and finally, in line 281 (conclusion section), it is declared that "A short period of cold treatment at 0 ℃ may increase the extension ability of the dominant leg"; Does this mean that the results may be different on the other leg? If the answer is affirmative, what have been the clear criteria to determine that this leg is the dominant one?
Response: Thank you very much. We have revised the title and manuscript to avoid possible confusion. This section has been changed to read as follows: This study further illustrates the effect of temperature of leg extension performance. A short period of cold treatment at 0℃ may increase the leg extension ability.
Reviewer 2 Report
Effect of cryotherapy temperature on the extension performance of the dominant leg in healthy adults
Journal: Biology
The aim of this study is to investigate the effect of 0℃ and -5℃ temperature cryotherapy on the dominant leg extension performance via measurement of mechanical power output. The authors hypothesized that, with a decrease in cryotherapy temperature, participants’ dominant leg extension performance would exhibit a different performance level of PP and PF that compared with a warmer cryotherapy temperature.
The approach of the study appears very original and the contents of the manuscript are quite interesting by his methodology and through the tools of quantification used.
The manuscript reads smoothly and is easy to understand. The aims, scope, and results of the study are clearly stated. I have very much enjoyed reading this paper. I find it interesting and clearly written, and satisfying also all the other publication criteria of the “Biology”. The study provides a very valuable addition to this line of research, and adds relevantly to the subject with additional original findings. I thus find that this paper definitively delivers results that will surely be of interest to the readership of the “Biology”. I recommend the publication of this interesting paper after the take into account the following references that can improve methodology and the discussion of this interesting study:
*Vitenet, M, Legrand, F., Bouchet, B., Bogard, F., Taiar, R, Polidori, G., Rapin, A., Boyer, FC. (2018). Whole body cryotherapy in fibromyalgia patients: Effects on pain and functional mobility, Annals of Physical and Rehabilitation Medicine, Volume 61, Supplement, July 2018, Pages e1-e2, https://doi.org/10.1016/j.rehab.2018.05.004.
Author Response
Comment 1. The aim of this study is to investigate the effect of 0℃ and -5℃ temperature cryotherapy on the dominant leg extension performance via measurement of mechanical power output. The authors hypothesized that, with a decrease in cryotherapy temperature, participants’ dominant leg extension performance would exhibit a different performance level of PP and PF that compared with a warmer cryotherapy temperature.
The approach of the study appears very original and the contents of the manuscript are quite interesting by his methodology and through the tools of quantification used.
The manuscript reads smoothly and is easy to understand. The aims, scope, and results of the study are clearly stated. I have very much enjoyed reading this paper. I find it interesting and clearly written, and satisfying also all the other publication criteria of the “Biology”. The study provides a very valuable addition to this line of research, and adds relevantly to the subject with additional original findings. I thus find that this paper definitively delivers results that will surely be of interest to the readership of the “Biology”. I recommend the publication of this interesting paper after the take into account the following references that can improve methodology and the discussion of this interesting study:
*Vitenet, M, Legrand, F., Bouchet, B., Bogard, F., Taiar, R, Polidori, G., Rapin, A., Boyer, FC. (2018). Whole body cryotherapy in fibromyalgia patients: Effects on pain and functional mobility, Annals of Physical and Rehabilitation Medicine, Volume 61, Supplement, July 2018, Pages e1-e2, https://doi.org/10.1016/j.rehab.2018.05.004.
Response: Thank you very much for your recognition and encouragement of our research, which greatly encourages our future in-depth research in this field. In fact, we have already carried out the following research. Thank you very much for the reference you recommended, which is of great reference significance for our research.
Reviewer 3 Report
Thnaks for the opportunity to review this interesting manuscript entitle "Effect of cryotherapy temperature on the extension perfor-2 mance of the dominant leg in healthy adults". I suggest the following changes in order to improve this manuscript:
- Abstract: please, add a clear conclusion at the end of the abstract.
- Also, study design is missing in the abstract.
- Introduction provides a complete state-of-the-art and is useful with clear aim and hypothesis.
- Methods: study design is missing. Thsi is an experimental study and needs to be registered at clincalTrials or similar databases. As a quasi-experimental study, the TIDieR criteria should be cited and you should affirm that these criteria have been followed. in addition, provide the checklist completed with page numbers as a non-published material.
- Why non-dominant leg was not included? Provide a justification in the introduction or methods..
- Reliability and validity coefficients should be added and cited for peak power and peak force. In addition, minimum detectable change may be very important.
- Sample size calculation is missing or with lacking information. This is a major issue.
- Statistical analysis is very simple. Were all data parametric? Justify the Paired-samples T test for all anamyses.
- A multivariate linear regression analysis in order to predict the main dependent variables based on demographic data would be useful.
- Effect sizes should be added.
- Why only before-after measurements?
- The sample size is too low in order to display your conclusions. Only two dependent variable were used. Non-dominant leg was not measured nor compared. Physical activity data such as International Physical Activity Questionanire (IPAQ) could have been useful to determine the physical activity level of the sample and predict the outcome measurements.
Author Response
Thanks for the opportunity to review this interesting manuscript entitle "Effect of cryotherapy temperature on the extension performance of the dominant leg in healthy adults". I suggest the following changes in order to improve this manuscript:
Comment 1. Abstract: please, add a clear conclusion at the end of the abstract.
Response: Thank you very much for your suggestion. We totally agree with you. We have added the following contents in the abstract according to your suggestion: “A short period of cold treatment at 0℃ may increase the leg extension ability. The 0℃ cold therapy for a short period of time is more suitable than -5℃ for athletes to recover during the interval of competition. Athlete and coach could choose an appropriate temperature to help increased the performance of physical level and recovery.”
Comment 2. Also, study design is missing in the abstract.
Response: Thank you very much. In fact, we have described the design of this study in the abstract section. The contents are as follows: “Thirty-one male participants from the University volunteered to join in two different temperature cryotherapy. The peak power and peak force of right leg extension performance of each participant was measured by Keiser at the moment of before and after cryotherapy respectively.”
Comment 3. Introduction provides a complete state-of-the-art and is useful with clear aim and hypothesis.
Response: Thank you very much for your positive comment.. We sincerely hope that this manuscript will meet the requirements for publication.
Comment 4. Methods: study design is missing. This is an experimental study and needs to be registered at clincalTrials or similar databases. As a quasi-experimental study, the TIDieR criteria should be cited and you should affirm that these criteria have been followed. in addition, provide the checklist completed with page numbers as a non-published material.
Why non-dominant leg was not included? Provide a justification in the introduction or methods..
Response: Thank you very much for your advice. In fact, the materials and equipment used in this study are up to standard. In addition, this experiment does not study the difference between dominant leg and non-dominant leg. In order to avoid confusion to readers, we have deleted the description of dominant leg from the manuscript. The title of this study already changed as follows: “Effect of cryotherapy temperature on the extension performance in healthy adults leg”. And we added several details in the Materials and Methods sections, such as: All participants were asked to measure the experimental index by the right leg.
Comment 5. Reliability and validity coefficients should be added and cited for peak power and peak force. In addition, minimum detectable change may be very important.
Response: Thank you very much for your suggestions. Authors have elaborated the contents related to PP and PF in the introduction and discussion section, and the research on PP and PF has been deeply carried out. In addition, we agree with your opinion and suggestion that the minimum detectable change will be carried out in our future research.
Comment 6. Sample size calculation is missing or with lacking information. This is a major issue.
Response: Thank you very much for your advice. We have added the result information about the sample size calculation in the manuscript, which is as follows: “The sample size was calculated as 31 (Effect size = 0.669, power = 0.95)”.
Comment 7. Statistical analysis is very simple. Were all data parametric? Justify the Paired-samples T test for all analyses.
Response: Thank you very much for your suggestion. All data were analyzed by paired sample T test. All the data were normalized through boxplot analysis, and any data that does not meet the normal test was eliminated.
Comment 8. A multivariate linear regression analysis in order to predict the main dependent variables based on demographic data would be useful.
Response: Thank you very much for your valuable advice, you put forward a novel method, we completely agree with your opinion. We are seriously considering applying the method you proposed in future research.
Comment 9. Effect sizes should be added.
Response: Thank you very much for your advice. Several key details, such as T value and degree of freedom, have been added in the paper according to the situation. In addition, we also added the Effect sizes information about sample sizes. Hope to meet your requirements.
Comment 10. Why only before-after measurements?
Response: Thank you very much. The purpose of this study was to explore the influence of cold therapy at different temperatures on the extension ability of healthy adults' lower limbs. Time factor was not considered in this study, so this study only included the measurement before and after cryotherapy. Further, we have preliminarily completed another study that looked at the influence of duration on the effect of cryotherapy (cold therapy).
Comment 11. The sample size is too low in order to display your conclusions.
Response: Thank you very much. We have calculated the sample size of this study, and the results show that when the sample size is 31, the results of this study can be supported. The specific content is as follows: total sample size: 31, effect size: 0.669, Power: 0.95.
Comment 12. Only two dependent variable were used. Non-dominant leg was not measured nor compared. Physical activity data such as International Physical Activity Questionanire (IPAQ) could have been useful to determine the physical activity level of the sample and predict the outcome measurements.
Response: Thank you very much for your valuable comments. We totally agree with your comment. In this study, the related indexes of PP and PF can evaluate the lower limb extension ability of the subjects. In future studies, we will integrate your suggestion and use the International Physical Activity Questionanire (IPAQ) to determine the physical activity level of the sample and predict the outcome measurements. We also include this point in our limitation should be considered in future studies as well.
Round 2
Reviewer 1 Report
The authors have modified the title, corrected the design, and adapted the conclusions.
All the considerations proposed in the review have been taken into consideration.
Thank you for your attention and congratulations on your work.
Reviewer 3 Report
Authors have responded most of my comments. Nevertheless, the study type (i.e. quasi-experimental study) should be added in abstract and in a study design subsection following the TIDieR criteria (these criteria should be cited and in the methods and named).
According to this design, this study should be registed in ClinicalTrials or other database as an experimental study.